# Characteristics and interpretation of subgroup analyses based on tumour characteristics in randomised trials testing target-specific anticancer drugs: design of a systematic survey

Stefan Schandelmaier ,[1,2] Andreas M Schmitt,[3] Amanda K Herbrand,[3] Dominik Glinz,[1] Hannah Ewald,[4] Matthias Briel,[1,2] Gordon H Guyatt,[2,5] Lars G Hemkens,[1] Benjamin Kasenda[3,6]

For numbered affiliations see end of article.

**Correspondence to**
Dr Stefan Schandelmaier;
s.schandelmaier@gmail.com

## ABSTRACT

**Background** Target-specific anticancer drugs are under rapid development. Little is known, however, about the risk of administering target-specific drugs to patients who have tumours with molecular alterations or other characteristics that can make the drug ineffective or even harmful. An increasing number of randomised clinical trials (RCTs) investigating target-specific anticancer drugs include subgroup analyses based on tumour characteristics. Such subgroup analyses have the potential to be more credible and influential than subgroup analyses based on traditional factors such as sex or tumour stage. In addition, they may more frequently lead to qualitative subgroup effects, that is, show benefit in one but harm in another subgroup of patients (eg, if the tumour characteristic makes the drug ineffective or even enhance tumour growth). If so, subgroup analyses based on tumour characteristics would be highly relevant for patient safety. The aim of this study is to systematically assess the frequency and characteristics of subgroup analyses based on tumour characteristics, the frequency of qualitative subgroup effects, their credibility, and the interpretations that investigators and guidelines developers report.

**Methods and analysis** We will perform a systematic survey of 433 RCTs testing the effect of target-specific anticancer drugs. Teams of methodologically trained investigators and oncologists will identify eligible studies, extract relevant data and assess the credibility of putative subgroup effects using a recently developed formal instrument. We will systematically assess how trial investigators interpret apparent subgroup effects based on tumour characteristics and the extent to which they influence subsequent practice guidelines. Our results will provide empirical data characterising an increasingly used type of subgroup analysis in cancer trials and its potential impact on precision medicine to predict benefit or harm.

**Ethics and dissemination** Formal ethical approval is not required for this study. We will disseminate the findings in a peer-reviewed and open-access journal publication.

## Strengths and limitations of this study

► We will use rigorous methodology including a systematic search for oncology trials published in leading journals, duplicate data extraction by a team involving both experienced methodologists and oncologists, transparent documentation including the collection of verbatim quotes, and use of a formal instrument for assessing the credibility of claimed subgroup effects.

► The systematic survey will specifically address subgroup claims based on tumour characteristics, which become increasingly relevant for decision making in an era of precision medicine.

► Potential limitations include a small number of eligible subgroup claims based on tumour characteristics, suboptimal reporting of identified subgroup claims and lack of subgroup analysis plans.

## INTRODUCTION

The increasing understanding of the biology of malignancies and the availability of new biotechnologies has led to a rapid development of anticancer drugs directed at molecular targets. The hope associated with a target-specific (or biomarker-driven) therapy is to maximise anticancer effects and minimise side effects. Prominent examples include BRAF inhibitors for melanoma,[1] tyrosine kinase inhibitors for patients with mutated epidermal growth factor receptor[2] (EGFR), or overexpression of the programmed death ligand-1 protein.[3]

Target-specific anticancer drugs are designed to directly inhibit tumour growth or enhance immunological antitumour response, by influencing a known—or at least partly understood—molecular mechanism. Typically, the targeted mechanism is complex

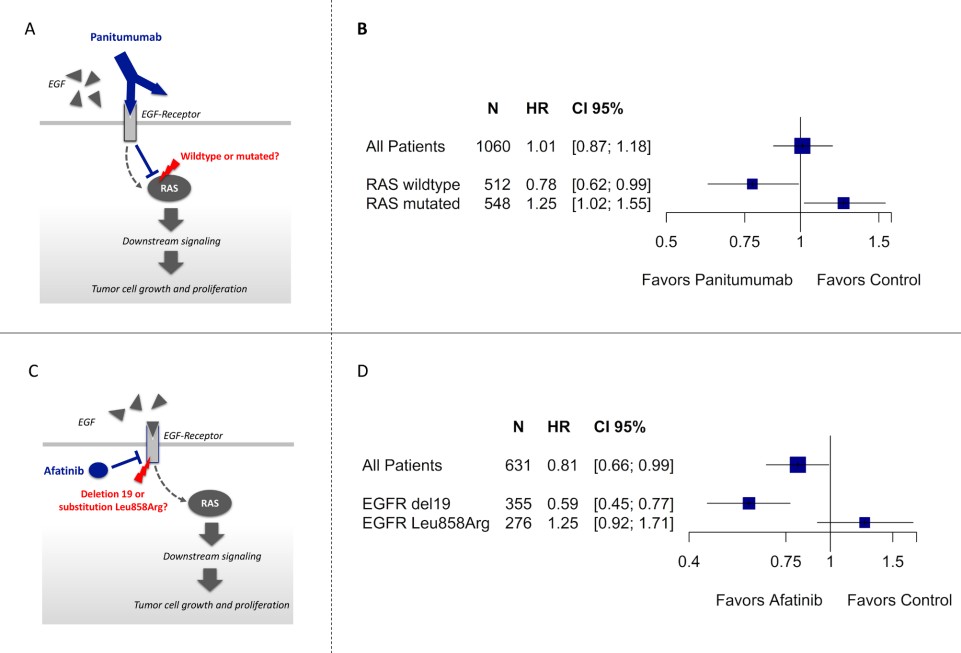

**Figure 1** (A) Mechanisms of action of panitumumab and molecular alteration of RAS protein. (B) Corresponding randomised clinical trial in patients with colorectal cancer; the trial reports a subgroup analysis for outcome survival suggesting benefit in patients with RAS wild-type tumours but harm in patients with RAS-mutated tumours.[5] (C) Mechanism of action of afatinib and molecular alteration of epidermal growth factor receptor (EGFR). (D) Corresponding randomised clinical trial in patients with lung cancer; the trial reports a subgroup analysis for outcome survival suggesting benefit in patients with EGFR deletion 19 but harm in patients with EGFR substitution Leu858Arg.[6]

and spans several steps starting with an interaction of the drug with the target molecule, followed by a signalling cascade, leading to endpoints relevant for tumour growth such as proliferation or apoptosis. Alterations of the molecules involved in this mechanism have the potential to modify the effect of the drug.

Anticancer treatments typically have side effectsand are judged acceptable under the assumption that the benefits will outweigh the side effects. Molecular alterations of the tumor could affect this net benefit and render the drug useless or even harmful for certain patients.

Investigators of randomised clinical trials (RCTs) increasingly use subgroup analyses to explore effect modifications by tumour characteristics. Those include subgroup analyses based on specific molecular alterations (eg, certain BRAF mutations), and also more unspecific tumour characteristics such as measures of mutation burden (ie, composite variables of several alterations), tumour grade, or histological subtype. A recent survey of cancer trials showed that 103 of 221 (47%) oncology trials published between 2011 and 2013 reported subgroup analyses based on biomarkers.[4]

For instance, an RCT in patients with colorectal cancer addressed the impact of panitumumab, a monoclonal EGFR antibody.[5] The downstream signalling pathway of panitumumab includes proteins encoded by the RAS gene family. Mutations in RAS genes might therefore modify the effect of panitumumab (figure 1A). A subgroup analysis of that RCT suggested that patients with wild-type RAS proteins have a significant survival

benefit (HR: 0.78, 95% CI: 0.62 to 0.99), whereas patients with a mutation in one of the RAS genes seemed to have worse survival when treated with panitumumab (HR: 1.25, 95% CI: 1.02 to 1.55, figure 1B). A statistical test of interaction suggested that chance may not satisfactorily explain the observed subgroup difference (interaction p value=0.01).

Another RCT addressed afatinib, which also targets the EGFR pathway (figure 1C), in patients with lung cancer.[6] The study included patients with a mutation in the EGFR gene, either deletion 19 or substitution Leu858Arg, genetic alterations that might modify the effect of afatinib. Patients with deletion 19 had a significant survival benefit (HR: 0.59, 95% CI: 0.45 to 0.77), whereas the effect in patients with substitution Leu858Arg suggested harm (HR: 1.25, 95% CI: 0.92 to 1.71, figure 1D).[6] Chance was an even less likely explanation of the apparent subgroup effect than in the prior example (p=0.0003).

These two examples suggest that target-specific agents may be highly beneficial in one but potentially harmful in another subgroup of patients. The authors of the two trials may have underemphasised the potential risk of giving target-specific drugs to the inferior subgroup. Rather than acknowledging the potential for serious harm, they concluded a 'lack of response'[5] or 'absence of effect'[6] in the inferior subgroups. Evidence users may be misled by such conclusions and erroneously assume safety when administering those drugs to patients in whom it may actually cause harm. Moreover, the neglect of possible harm may result in a missed opportunity for

further research investigating the reasons and mechanisms for the potential harmful effects.

In contrast, findings suggested by subgroup analyses have generally low credibility. The majority of claimed subgroup effects are not confirmed in subsequent research[7] and even dramatic-looking subgroup effects that are subsequently found to be spurious are not rare.[8 9] There is vast empirical evidence documenting the high risk of spurious subgroup effects,[4 7 10–25] the main reasons being weak hypotheses, multiple testing, selective reporting and inappropriate statistical analysis.

Some have suggested that subgroup analyses based on tumour characteristics, compared with traditional subgroup analyses (eg, by age or sex), might lead to more credible findings:[26] first, subgroup effects based on tumor characteristics may have a high biological plausibility (ie, a clear a priori hypothesis); second, the side effects typically associated with anticancer drugs may—in the absence of benefit—actually cause harm and corresponding subgroup effects[27 28] may therefore be more likely to be large enough to be detected by statistical tests.[27 28] For instance, the two examples suggested large qualitative subgroup effects (harmful in one subgroup and beneficial in another) with p values suggesting chance as an unlikely explanation.[5 6]

No empirical study is available that addresses subgroup effects based on tumour characteristics in target-specific anticancer therapy. Therefore, we will perform a systematic survey of RCTs to determine if the presented examples are isolated instances or a generalisable phenomenon.

Specific study objectives are

1. To assess, in RCTs investigating target-specific anticancer drugs, the frequency of
   – subgroup analyses based on tumour characteristics.
   – claims about subgroup effects based on tumour characteristics made by trial investigators.
   – qualitative subgroup effects based on tumour characteristics, that is, suggesting benefit in one and harm in another subgroup.
2. To assess the credibility of claimed subgroup effects based on tumour characteristics.
3. To assess, for claimed subgroup effects, how trial investigators or guideline developers interpreted the results (eg, mentioned the potential for harm) and how these interpretations relate to the credibility assessment (eg, it would be problematic to emphasise subgroup effects with low credibility or not highlight subgroup effects with high credibility).

## METHODS AND ANALYSIS
### Sample of randomised trials
We will include publications of parallel-group RCTs that (1) enrol patients with a malignant disease; and (2) include a comparison to investigate the effect of a target-specific anticancer drug. We will use an inclusive definition of 'target-specific anticancer drug' and consider any drug with a defined mode of pharmacodynamic action,

for example, tyrosine kinase inhibitors targeting specific proteins resulting from mutations, all monoclonal antibodies targeting specific proteins or antihormonal drugs (eg, oestrogen or androgen receptor antagonists and/or agonists)

### Search strategy
Based on a sensitive search strategy that we developed together with a medical information specialist, we searched PubMed for eligible RCTs published since 2014 in the nine oncology and four general medical journals with the highest impact factor in the year 2016. We used free text and medical subject headings for terms related to cancer and combined them with journal indexations and a validated filter for RCTs[29] (online supplementary appendix A). Teams of two investigators, working independently and in duplicate, will screen abstracts for potentially eligible studies, for which we will acquire full texts. A database that our group has established for a related project will provide an initial list of target-specific drugs that we will extend during the screening.[30] Teams of two investigators, one of whom will be an oncologist, will assess final eligibility and, if needed, resolve disagreements by discussion or, if needed, third-party adjudication. The teams will also independently identify reported tumour characteristics used for subgroup analyses. For RCTs that claim one or more subgroup effects based on tumour characteristics, we will screen references and trial registries to identify corresponding study protocols, which are relevant for the credibility assessment. For each subgroup claim, we will search corresponding statements in the most current version of clinical guidelines from the American Society of Clinical Oncology, European Society of Medical Oncology and UpToDate.com.

### Sample size
Our search identified 1119 potentially relevant abstracts of which we already acquired and screened the full texts. Of those, 433 proved eligible RCTs testing a target-specific anticancer therapy and reporting one or more subgroup analysis. Based on previous research,[4] we estimate that approximately 90 of them will report a subgroup analysis based on a tumour characteristic. Of those, we estimate that approximately 35 will lead to claimed subgroup effects.[11] This number of RCTs and claims should be sufficient to achieve the study objectives.

### Data extraction
We will develop and standardise extraction forms with detailed explanations for each data point. Teams of two methodologically trained investigators will extract data from publications and associated protocols and resolve disagreements by discussion.

We will extract the following information for each trial: population; interventions; primary outcome(s), the overall treatment effect including CIs; whether the trial reported any subgroup analyses (or analyses of effect modification based on a continuous variable) in the

main paper, appendices or in secondary publications; number of effect modifiers (ie, subgrouping variables) type of effect modifiers (categorised by oncologists into (1) tumour characteristics of interest defined as genetic alterations, grade, subtype, or other molecular or histological characteristics; (2) tumour characteristics not of interest such as staging, location, size, or other macroscopic or radiological characteristics; or (3) other effect modifiers such as age or sex), subgroup hypotheses (if reported, eg, a rationale explaining how a genetic alteration might diminish the effect of a target-specific drug), methods used for subgroup analyses (eg, interaction term in cox model, forest plot and multivariable analysis of effect modification), numerical results of subgroup analyses (eg, point estimates and CIs in each subgroup, p value from a test of interaction), and whether authors make no, a weak, a moderate or a strong claim of effect modification.

### Assessing the strength of subgroup claims

To assess the strength of reported subgroup claims, we will use predefined criteria developed by Sun *et al.*[11] The criteria address where the authors presented the claim (abstract, conclusion of the abstract and discussion), whether they used descriptive words to strengthen (eg, using words such as 'particular') or soften (eg, using words such as 'might') the claim, obvious notes of caution (eg, 'but not significant') or indicated that results need to be explored in the future. Based on the assessment, the claims will then be categorised as (1) no claim; (2) suggestion of a possible effect modification; (3) claim of a likely effect modification; (4) or a strong claim of effect modification. Two reviewers will independently assess the presence and strength of claims, collect supporting verbatim quotes and resolve discrepancies by discussion, if necessary, with the help of a third reviewer.

### Assessing the credibility of claimed subgroup effects

To assess the credibility of claimed effect modification (based on tumour characteristics), we will apply a recently developed instrument for assessing the credibility of effect modification analyses (ICEMAN).[31] In the version for RCTs, the instrument provides five core questions:

1. Was the direction of the effect modification correctly hypothesised a priori?
2. Was the effect modification supported by prior evidence?
3. Does a test for interaction suggest that chance is an unlikely explanation of the apparent effect modification?
4. Did the authors test only a small number of effect modifiers or consider the number in their statistical analysis?
5. If the effect modifier is a continuous variable, were arbitrary cut-points avoided?

The instrument provides four predefined responses for each question and an overall credibility rating on a continuous scale ranging from very low to high credibility. Teams of two methodologically trained investigators

will independently apply ICEMAN to claimed subgroup effects based on tumour characteristics and resolve disagreements by discussion. Items 1, 2, 4 and potentially 5 require a study protocol for optimal assessment. Therefore, for each study reporting a claim, we will search for corresponding study protocols by screening references and trial registry entries.

### Statistical analysis

We will use descriptive statistics (frequency, proportions and distribution) to present trial characteristics, including the number and type of subgroup analyses reported per trial, number and strength of claims made, and whether a subgroup claim is qualitative. A qualitative subgroup claim means that (a) authors claim a subgroup difference plus (b) the point estimates suggest benefit in one harm in another subgroup. (Note that a situation in which one subgroup suggests an HR of 0.99 and another subgroup, an HR of 1.01 would not count due to the lack of a claim. Typically, authors make a claim only if point estimates differ substantially or a test of interaction suggests a p value of ≤0.05.)

For all individual subgroup claims, we will present the type of effect modifier (tumour characteristics vs other), the numerical results, the credibility rating and the interpretations presented by primary study authors and/or guidelines. We will also create forest plots showing, for all claimed subgroup effects, point estimates and associated CIs within subgroups, differentiating MA-based and other subgroup effects. We will clearly indicate if a trial reports more than one subgroup claim.

A key criterion for the credibility assessment is the result of an interaction test (usually a p value). We know from previous empirical studies that most trials reporting subgroup effects do not include the results of a test of interaction.[4 7 11] We will report the proportion of studies that report an appropriate test of interaction, present sufficient data to calculate a p value of interaction, or fail to present a p value and sufficient data to calculate a p value of interaction. Because here we are interested in the credibility of subgroup claims rather than reporting quality, we will perform missing interaction tests whenever possible. For instance, if subgroup-specific estimates and CIs are reported, then we will calculate the standard errors and approximate the test of interaction by performing a t-test. We will transparently report when we imputed missing data points.

### Patient and public involvement

Patients and/or the public were not involved in the design, or conduct, or reporting, or dissemination plans of this research.

### DISCUSSION

This study will provide insights into the effectiveness and safety of target-specific anticancer drugs—in particular, in patients subgroups with potentially relevant tumour

characteristics. Our study will clarify as yet unresolved issues of whether subgroup analyses based on tumour characteristics performed as part of RCT reports deliver useful results and whether trial investigators and guideline developers draw appropriate conclusions.

Several previous studies have systematically assessed the credibility of subgroup effects,[4 7 11] but none has focused on tumour characteristics in the context of target-specific anticancer drugs. A strength of our study will be the use of formally developed ICEMAN credibility instrument.[31 32] Previous studies that investigated the credibility of subgroup claims have used ad hoc criteria or checklists with important limitations including non-transparent development process, criteria based on individual expert opinion rather than consensus, vagueness in criteria and lack of user testing.[33] ICEMAN is a new instrument based on a systematic survey of the relevant methodological literature, an explicit measurement concept, a formal instrument development process together with a consensus panel consisting of leading experts, and systematic user testing.[31 32] Another strength of the study is that we will focus on subgroup claims that are more likely to be based on causal hypotheses and thus likely more credible than subgroup claims based on sex, region or other characteristics for which there is usually little reason to suspect effect modification.[34 35]

We do not anticipate any serious feasibility problems. In previous empirical studies, investigators have successfully used similar methods, including systematic surveys of subgroup analyses in cancer trials,[4] assessing the strength of subgroup claims,[7 11 12] and qualitative analysis of clinical practice guidelines.[30] We anticipate some difficulties in rating the credibility of subgroup claims related to reporting quality. For instance, some trials will not provide a published protocol or insufficiently report the number of effect modifiers tested or numerical results.[18] As implemented in the ICEMAN instrument, we will take the position that insufficient reporting is a reason for lower credibility. Because rating the credibility of subgroup analysis and appropriateness of interpretation inevitably involves judgement, different investigators may come to different conclusions. Therefore, in order to reduce the variability of judgements, two investigators will make independent assessments and come to a consensus by discussion. In addition, we will provide verbatim quotes to support our judgements in a transparent way.

A limitation of quantitative results such as the frequency of subgroup claims based on tumour characteristics in cancer trials will be a high risk of reporting and publication bias. We will therefore make clear that our results will reflect what is reported and not necessarily how often authors consider those analyses. This potential limitation, however, will not weaken inferences made for individual RCTs. Another potential limitation is that, although we know from previous research that subgroup analyses based on biomarkers are included in almost half of oncology trials,[4] we have no specific information about the frequency of subgroup analyses based on tumour

characteristics. It might be possible that we will identify only a small number of relevant claims based on tumour characteristics and corresponding guideline statements.

In summary, this systematic survey of RCTs will address characteristics, frequency, credibility and impact of effect modification analysis based on tumour characteristics. If our findings support the alarming hypothesis that the risk of harm caused by target-based drugs might be systematically underemphasised, then our results will help to increase the awareness of the issue among patients, oncologists, trial investigators, journal editors, and guideline developers. The results may trigger a re-evaluation of existing trials, inform planning of future trials and influence how trial investigators and guideline developers interpret MA-based subgroup analyses in the context of target-specific anticancer therapy.

## ETHICS AND DISSEMINATION
Formal ethical approval is not required for this study that will be based on published information only. We will disseminate the findings in a peer-reviewed and open-access journal publication.

**Author affiliations**
[1]Institute for Clinical Epidemiology and Biostatistics, Department of Clinical Research, University Hospital and University of Basel, Basel, Switzerland
[2]Department of Health Research Methods, Evidence, and Impact, McMaster University, Hamilton, Ontario, Canada
[3]Department of Medical Oncology, University Hospital Basel, Basel, Switzerland
[4]University Medical Library, University of Basel, Basel, Switzerland
[5]Department of Medicine, McMaster University, Hamilton, Ontario, Canada
[6]Research and Development, iOMEDICO AG, Freiburg, Germany

**Acknowledgements** The authors thank Kuebra Oezoglu for acquiring the full texts.

**Contributors** SS, LH, GG, MB and BK: conceptualised the study. HE, DG and BK: developed the search strategy and screened titles and abstracts. SS, BK, AKH and AMS: screened full texts. All authors discussed and edited drafts of the manuscript and read and approved the final version.

**Funding** The work was supported by a personal grant to StS from the Swiss National Science Foundation (project number 180830).

**Competing interests** BK reports consultant activities for Roche and Siemens, research Grants from Roche/AbbVie and travel support from Riemser, AbbVie and Amgen—all not related to the project described herein.

**Patient consent for publication** Not required.

**Provenance and peer review** Not commissioned; externally peer reviewed.

**ORCID iD**
Stefan Schandelmaier http://orcid.org/0000-0002-8429-0337

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
