## [Reviewer comments · BMJ Open]

ARTICLE DETAILS

TITLE (PROVISIONAL)	Characteristics and interpretation of subgroup analyses based on tumor characteristics in randomized trials testing target-specific anti-cancer drugs: design of a systematic survey
AUTHORS	Schandelmaier, Stefan; Schmitt, Andreas; Herbrand, Amanda; Glinz, Dominik; Ewald, Hannah; Briel, Matthias; Guyatt, Gordon; Hemkens, Lars; Kasenda, Benjamin

VERSION 1 – REVIEW

REVIEWER	Umberto Malapelle University of Naples, Federico II
REVIEW RETURNED	25-Nov-2019

GENERAL COMMENTS	The manuscript entitled " Characteristics and interpretation of subgroup analyses based on molecular alterations in randomized trials testing target-specific anti-cancer therapy: design of a systematic survey" by Schandelmaier et al is well written and suitable for publication after minor revisions:  -in the section methods, please, could the Authors report why ICEMAN modification analysis system was adopted in the study? could the authors report its limitation and point of strenght? Could the authors evaluate if other analysis method may be considered for this manuscript? - in the section statysical analysis, could the authors better argue this concept "Therefore, whenever possible, we will perform missing interaction tests. For instance, if subgroup specific estimates and confidence intervals are reported, we will calculate the standard errors and approximate the test of interaction by performing a t-test. We will transparently report when we imputed missing data points. " - in the section discussion ,could the authors think if a multivariate statysical anlysis may be performed to corroborate statysical analysis?
--

REVIEWER	Martyna Filipka Institut d'Investigació Germans Trias i Pujol (IGTP), Spain
REVIEW RETURNED	16-Dec-2019

GENERAL COMMENTS	I revised with a great pleasure the submitted protocol design. Authors have addressed a very important question of subgroup analysis in randomized clinical trials. It further leads to questions that goes beyond, namely to the occurence of co-alterations in patients subgroups, I do understand however that this is a separate question. Maybe it would be valuable to consider for a next study design. It has been proved not only by our group experience that other alterations (not necessarily of the "target") may affect patient's
---

	benefit.
REVIEWER	Dr. Christine Baulig MSc. Institute of Medical Biometry and Epidemiology Faculty of Health University of Witten/Herdecke Alfred Herrhausen-Str. 50 58448 Witten Germany
REVIEW RETURNED	27-Jan-2020
GENERAL COMMENTS	A very interesting question, one can look forward to the results. Please make sure that the entire data collection of the two investigators took place independently.

VERSION 1 – AUTHOR RESPONSE

Reviewer 1

Reviewer Name: Umberto Malapelle

Institution and Country: University of Naples, Federico II

Please state any competing interests or state 'None declared': No competing interest

Please leave your comments for the authors below

The manuscript entitled " Characteristics and interpretation of subgroup analyses based on molecular alterations in randomized trials testing target-specific anti-cancer therapy: design of a systematic survey" by Schandelmaier et al is well written and suitable for publication after minor revisions:

Thanks

-in the section methods, please, could the Authors report why ICEMAN modification analysis system was adopted in the study? could the authors report its limitation and point of strength? Could the authors evaluate if other analysis method may be considered for this manuscript?

Response: The revised manuscript includes the following, extended justification for the use of ICEMAN in the discussion section:

“Several previous studies have systematically assessed the credibility of subgroup effects,^{4,7,11} but none has focused on tumor characteristics in the context of target-specific anti-cancer drugs. A strength of our study will be the use of formally developed ICEMAN credibility instrument.^{30,32} Previous studies that investigated the credibility of subgroup claims have used ad-hoc criteria or checklists with important limitations including non-transparent development process, criteria based on individual expert opinion rather than consensus, vagueness in criteria, and lack of user-testing.³¹ ICEMAN is a new instrument based on a systematic survey of the relevant methodological literature, an explicit measurement concept, a formal instrument development process together with a consensus panel consisting of leading experts, and systematic user-testing.^{30,32}”

The main limitation of ICEMAN is its dependency on the reporting quality of the subgroup claims, which we state in the following:

“We anticipate some difficulties in rating the credibility of subgroup claims related to reporting quality. For instance, some trials will not provide a published protocol or insufficiently report the number of effect modifiers tested or numerical results.¹⁸”

We are not aware of another method for assessing the credibility of subgroup claims apart from the new ICEMAN instrument and the mentioned previous checklists.

- in the section statistical analysis, could the authors better argue this concept "Therefore,

whenever possible, we will perform missing interaction tests. For instance, if subgroup specific estimates and confidence intervals are reported, we will calculate the standard errors and approximate the test of interaction by performing a t-test. We will transparently report when we imputed missing data points."

Response: We revised and extended the corresponding section as follows:

"A key criterion for the credibility assessment is the result of an interaction test (usually a p-value). We know from previous empirical studies that most trials reporting subgroup effects do not include the results of a test of interaction.^{4,7,11} We will report the proportion of studies that report an appropriate test of interaction, present sufficient data to calculate a p-value of interaction, or fail to present a p-value or sufficient data to calculate p-value of interaction. Because here we are interested in the credibility of subgroup claims rather than reporting quality, we will perform missing interaction tests whenever possible. For instance, if subgroup specific estimates and confidence intervals are reported, we will calculate the standard errors and approximate the test of interaction by performing a t-test. We will transparently report when we imputed missing data points."

- in the section discussion, could the authors think if a multivariate statistical analysis may be performed to corroborate statistical analysis?

Response: Multivariate analysis (i.e. analysis of multiple outcomes in a common model) does not seem relevant for our analyses. We are not planning to examine associations between variables. We deleted the possibly misleading statement "qualitative and quantitative methods" from the abstract.

Reviewer 2

Reviewer Name: Martyna Filipka

Institution and Country: Institut d'Investigació Germans Trias i Pujol (IGTP), Spain

Please state any competing interests or state 'None declared': None declared

Please leave your comments for the authors below

I revised with a great pleasure the submitted protocol design. Authors have addressed a very important question of subgroup analysis in randomized clinical trials.

Thanks

It further leads to questions that goes beyond, namely to the occurrence of co-alterations in patients subgroups, I do understand however that this is a separate question. Maybe it would be valuable to consider for a next study design. It has been proved not only by our group experience that other alterations (not necessarily of the "target") may affect patient's benefit.

Response: Thanks for this suggestion that we discussed in our team. We think you made a good point that different types of alterations can interact with the effectiveness of the target specific drug. In addition, molecular alteration can manifest itself in other tumor characteristics such as histological subtype. Therefore, we decided to be more inclusive and also include other tumor characteristics. We included the following in the revised protocol:

"We will extract the following information for each trial; ... ; type of effect modifiers (categorized by oncologists into: (1) tumor characteristics of interest defined as genetic alterations, grade, subtype, or other molecular or histological characteristics; (2) tumor characteristics not of interest such as staging, location, size, or other macroscopic or radiologic characteristics; or (3) other effect modifiers such as age or sex)"

We replaced "molecular alterations (MAs)" by "tumor characteristics" throughout.

Reviewer: 3

Reviewer Name: Dr. Christine Baulig MSc.

Institution and Country:

Institute of Medical Biometry and Epidemiology

Faculty of Health

University of Witten/Herdecke

Alfred Herrhausen-Str. 50

58448 Witten

Germany

Please state any competing interests or state 'None declared': None declared

Please leave your comments for the authors below

A very interesting question, one can look forward to the results.

Thanks

Please make sure that the entire data collection of the two investigators took place independently.

We added the following: "Teams of two investigators, working independently and in duplicate, ..."

Other changes:

In addition to broadening the definition of "molecular alterations" to "tumor characteristics", we also slightly broadened the definition of the sample. Instead of including only RCTs that report survival outcomes, we will now also include RCTs that include other types of outcomes (e.g. quality of life). We felt that the restriction was not necessary because most RCTs will report on survival outcomes anyway.

We made a number of minor corrections and edits to improve clarity.

VERSION 2 – REVIEW

REVIEWER	Malapelle Umberto Department of Public Health, University of Naples Federico II
REVIEW RETURNED	04-Mar-2020
GENERAL COMMENTS	The manuscript is suitable for publication without any modifications.